# Optimization of the Active Layer P3HT:PCBM for Organic Solar Cell

**Mohamed Shaban [1], Mohamed Benghanem [1,*]** **, Abdullah Almohammedi [1] and Mohamed Rabia [2,3]**

[1] Physics Department, Faculty of Science, Islamic University of Madinah, P.O. Box 170, Madinah 42351, Saudi Arabia; mssfadel@yahoo.com (M.S.); ard.almohammedi@hotmail.com (A.A.)

[2] Nanophotonics and Applications Laboratory, Physics Department, Faculty of Science, Beni-Suef University, Beni-Suef 62514, Egypt; mohamedchem@science.bsu.edu.eg

[3] Polymer Research Laboratory, Chemistry Department, Faculty of Science, Beni-Suef University, Beni-Suef 62514, Egypt

* Correspondence: benghanem_mohamed@yahoo.fr; Tel.: +966-507-346-783

**Abstract:** ITO/PEDOT:PSS/P3HT:PC$_{60}$BM/Mg-Al organic solar cells (OSCs) were fabricated depending on optimization of Poly(3-hexylthiophene-2,5-diyl) (P3HT) and phenyl-C61-Butyric-Acid-Methyl Ester (PC$_{60}$BM). The optimization of the active layer, P3HT:PC$_{60}$BM, was carried out under different spin frequencies coating from 900 to 3000 rpm. The post-production annealing temperature of all prepared OSC was studied from 130 to 190 °C. The holes transport layer, poly(3,4-ethylene dioxythiophene) polystyrene sulfonate (PEDOT:PSS), was prepared under constant conditions of 3000 rpm for 35 s, and annealing temperature 178 °C for 15 min. From our study, the optimum conditions for P3HT:PC$_{60}$BM were spin coating of 3000 rpm, and annealing temperature of 160 °C for 5 min. The optimum J-V parameters values for the prepared OSC were $J_{SC}$ = 12.01 mA/cm$^2$, $V_{OC}$ = 660 mV, *FF* = 59%, *PCE* = 4.65%, and *EQE* = 61%. A complete OSC with acceptable efficiency was designed using simple and low-cost techniques that may be utilized in the industry. Furthermore, the cost of the synthesized solar cell is projected to be around 1 \$/cm$^2$, with the goal of lowering the cost and increasing efficiency in the future by incorporating more commercial nanostructured electron/hole transport components.

**Keywords:** organic solar cells polymers; annealing temperature; external quantum efficiency; parasitic resistances

## 1. Introduction

The organic solar cell (OSC) is a new generation of photovoltaic devices. Due to its simpler and cheaper production methods, OSC has a smart future. Researchers are doing their best to maximize the OSC's properties and to surpass the limits of low power conversion efficiency (*PCE*) and fill factor (*FF*). For increasing the efficiency of the prepared OSC, the photon to electron conversion process must be optimized. This was carried out by using new materials with high activity, good band-gap, and good recombination processes of the charge carrier [1–5]. Researchers are doing their best to reach the optimum values for open-circuit voltage ($V_{OC}$) and short-circuit current density ($J_{SC}$) [6,7]. Active layer morphology, material structures, concentrations, and annealing temperature are the general parameters relevant to $V_{OC}$ and $J_{SC}$.

The annealing temperature is an important parameter for the enhancement of the *PCE* of the prepared organic solar cell. Through the annealing process, the morphology of the active layer is reshaped. This process affects the activity of the active layer, and in general, affects all the OSC. Recent work was presented about the impact of blend compositions and annealing temperatures on the optical characteristics of P3HT:PCBM-based OSC [8]. There were some other studies carried out to improve the *PCE* through the post-production annealing process for the prepared OSC [9,10]. Padinger et al. [11] obtained a *PCE* of

3.5% for ITO/PEDOT:PSS/P3HT:PCBM/LiF/Al OSC at a post-production annealing temperature of 75 °C for 4 min. Reyes et al. [12] optimized the annealing temperature and time. The maximum $J_{SC}$ was 11.1 mA/cm$^2$ at 150 °C for 5 min due to the improvement in the film crystallinity. Li et al. [13] reported a better improvement in *PCE* of OSC after post-production annealing due to the improvement of the morphology and the working of the cathode as a barrier. The polymer and the cathode were smoother in contact, resulting in a better set of charges.

In addition, the enhancement in the morphology of the active layer due to annealing optimization causes more enhancements in the optical properties of the active layers. Erb et al. [14] studied the relationship between optical properties and crystallinity of P3HT:PCBM. The isolated PCBM molecules were dispersed into larger aggregates after annealing, while P3HT aggregates were transformed into P3HT crystallites in these PCBM-free regions at the same time. The *PCE* enhancement of the annealed P3HT:PCBM OSC was due to the PCBM cluster's improved electron transportation and increased P3HT absorption.

Moreover, few studies tried to enhance the *PCE* of OSCs through annealing the active layers only. Chen et al. [10] obtained *PCE* of 3.70% for ITO/PEDOT:PSS/P3HT:PCBM/Al OSC by annealing the active layer for 5 min at 150 °C.

In addition, the organic layers are influenced by environmental conditions such as temperature, oxygen, moisture content, and light exposure [15–20]. Lin et al. [21] stated that the low-temperature drying of the P3HT:PCBM active layer of OSCs improved their thermal stability and *PCE*. Matthew et al. [22] studied P3HT and its blend and confirmed that the OSC was stable up to 1000 h when irradiated in an inert atmosphere, but when irradiated in the atmospheric environment, the stability decreased to 700 h. Maceu et al. [23] concluded that P3HTT aging was responsible for the oxidation of the alkyl side chain and the sulfur atom in the thiophene ring.

The superior features and optimum blend of P3HT and PCBM promote dense research on OSCs based on P3HT:PCBM [24].

The *PCE* of the prepared solar cell depends on the composite ratio of P3HT:PCBM. Padinger et al. [25] and Schilinsky et al. [26] analyzed the P3HT:PCBM composition ratios of 1:2 and 1:3 and concluded that the *PCE* values were 2.8% and 3.5%, respectively. At the 1:1 ratio composition, Huang et al. [27] studied the balanced electron and hole mobilities, and they figured that this gives a more orderly blend structure. Li et al. [28] used the 1,2-dichlorobenzene (oDCB) solvent for P3HT:PCBM and achieved 4.4% *PCE*. They varied the solvent evaporation periods to control the growth rate and hence reported the *PCE* values. They decided that slow-grown layers perform better than fast-grown layers. Supriyanto et al. prepared P3HT:PCBM variation of mass ratio but the *PCE* efficiency was not increased over 0.01% [29]. Chi et al. prepared P3HT:PCBM with an additional PCBM layer between the active layer and the cathode, in which the efficiency reached 4.24% [30]. Chang et al. fabricated P3HT:PCBM incorporated with silicon nanoparticles as a photoactive layer in more efficient organic photovoltaic devices, but he reached a maximum efficiency of 3.38% [31]. Khairulaman et al. improved the performance of inverted type organic solar cells using copper iodide-doped P3HT:PCBM as the active layer, but the maximum *PCE* was 2.4% [32].

The PEDOT:PSS is applied as a hole transport layer (HTL) in the prepared OSCs. It possesses a 5 eV working function and triggers the barrier of holes injection. In addition to that, PEDOT:PSS plenaries the rough ITO surfaces and thus avoids all local short-circuiting. Moreover, its high transmission and conductance are beneficial for the application of solar cells [33]. Therefore, it is important to design P3HT:PC$_{60}$BM-based OSC with *PCE* close to 5% through optimizing the fabrication parameters such as the spinning velocity and annealing temperature and testing different cathode metals.

In this study, ITO/PEDOT:PSS/P3HT:PC$_{60}$BM/Mg-Al OSCs were fabricated by varying different parameters including post-production temperature of OSC, spin coating frequency of the active layer, and the use of Al or Mg-Al as a cathode. The hole trans-



port layer was PEDOT:PSS, which was prepared under the same conditions for all OSCs. P3HT:PC$_{60}$BM concentration was 2 wt.%, and the spin coating was varied from 900 to 3000 rpm. The post-production temperature was varied from 130 to 190 °C. The optimum J-V parameters values for the prepared OSC were determined under different preparation conditions. Moreover, the key performance indicators are obtained and optimized.

## 2. Experimental Part

### 2.1. Materials

P3HT (698997-5G), PC$_{60}$BM (Solenne b.v, Groningen, The Netherlands, 99.5%), and ITO glasses were bought from Sigma Aldrich, Burlington, VT, USA. PEDOT:PSS (Clevios, Heraeus Epurio, Hanau, Germany), and Magnesium and Aluminium metals were purchased from VWR company, Ismaning, Germany.

### 2.2. Solution Preparation

The hole transport layer (PEDOT:PSS (Clevios, Heraeus Epurio, Hanau, Germany)) is used directly without any dilution, in which a filter of 0.45 μm is used. The active and electron transport layer, P3HT:PC$_{60}$BM, is prepared 3:2 (1.8 wt.%:1.2 wt.%) in which 72 mg P3HT and 48 mg in 6 mL chlorobenzene are dissolved for 15 days in the glovebox using an additive solution of the peprazine (1 wt.%).

### 2.3. OSC Fabrication

The OSC device preparation was carried out under different parameters; varying the spin coating for P3HT:PC$_{60}$BM active layer (900 to 3000 rpm), post-production annealing temperature for all the prepaid solar cells, and changing the cathode metals.

ITO (2.5 cm × 2.5 cm) was cleaned using water soap, toluene, acetone, and iso-propanol, in which the cleaning was carried out using ultrasonic for 15 min for each solution. PEDOT:PSS was coated on ITO using spin coater working at 3000 rpm for 15 s, then annealed at 178 °C for 15 min in the air environment. After the annealing process, the ITO/PEDOT:PSS was transferred directly inside the glove box for drying. P3HT:PC$_{60}$BM solutions were deposited on PEDOT:PSS at a spinning velocity varied from 900 to 3000 rpm. The thickness of P3HT:PC$_{60}$BM was ~290 nm at 900 rpm and reduced to ~160 nm at 3000 rpm, in a good approximation with the reciprocal square root of the spin speed. Finally, the cathode metals (Al or Mg-Al) coated the previous layers. The evaporation of the metals was carried out using the physical evaporation process at $2.8 \times 10^{-6}$ bar with a total thickness of 463 nm. The post-annealing process was studied on the prepared OSC. The annealing temperatures from 130 to 190 °C were carried out on the prepared OSC for 5 min inside the glove box.

### 2.4. OSC Characterization and Experimental Measurements

The analyses were carried out for the prepared active layers, P3HT:PC$_{60}$BM, and all the OSCs. For the produced P3HT:PC$_{60}$BM active layer, optical spectra of absorbance, transmittance, and reflectance were measured using SPEC ORD ANALYTIC JENA AG spectrophotometer. The wavelength ranged from 300 to 1100 nm. The optical spectra were measured at room temperature. The electrical analyses were carried out for all prepared OSCs; J-V in light (under solar simulator, Model CT100AAA, PET Inc., Moorpark, CA, USA) and dark (Computer-controlled source, Keithley 2400 AM1.5 illumination, (Keithley, Solon, OH, USA) Solar Light, XPS 400, (Solar Light Inc., Glenside, PA, USA) [33]. The average data points, *FF*, $R_P$, $R_S$, $J_{SC}$, $V_{OC}$, and *PCE*, were gathered from four-time repetitions using four solar cells prepared under identical conditions to obtain more precise measurements and findings. This means that we constructed four solar cells under identical conditions and averaged the results; this technique of research yields more precise measurements and outcomes.

EQE measurements were recorded under the illumination of a monochromatized halogen lamp. The incident light beam was chopped with a mechanical chopper and the photocurrent was detected using a lock-in-amplifier.

## 3. Results and Discussion

The optical behaviors of the prepared active layer, P3HT:PC$_{60}$BM, were studied under different annealing temperature for the prepared ITO/PEDOT:PSS/P3HT:PC$_{60}$BM/Al OSC. The concentration of the P3HT:PC$_{60}$BM solution was 2 wt.% and the spin coating was carried out at 1500 rpm for 60 s. Figure 1a represents the reflectance and transmittance spectra for the active layer, while Figure 1b shows the absorbance spectra. The main absorption features are observed in Figure 1b. The absorption shoulder around ~332 nm is associated with PCBM, but the main peak at higher wavelengths is associated with P3HT absorption. The peak at ~487 nm is ascribed to band-to-band transitions inferred from π–π* transitions between the allowed highest occupied molecular orbital (HOMO) and the lowest unoccupied molecular orbital (LUMO) of P3HT:PC$_{60}$BM. At high annealing temperature (>160 °C), this peak displayed a small blue shift, which might be attributed to the interruption of the structure and the orientation for the P3HT chain ordering due to the heat treatment. Absorption features (S0, S1, S2) corresponding to P3HT excitonic absorption with the participation of Frenkel excitons were found at ~602, 550, and 510 nm, respectively. These excitonic absorptions correspond to the formation of a singlet exciton, one exciton, and one phonon, and one exciton plus two phonons, respectively. The absorption at 602 nm refers to P3HT inter-chain stacking, signifying an improvement in chain ordering whilst the absorption at 550 nm is allocated to the extended conjugated polymer. From Figure 1a,b, the active layer works well in the UV and Vis light region, especially from 300 to 650 nm. At 140 °C, the transmission is maximum in the IR region and minimum in the UV/Vis region. In other words, a reversal behavior is observed between IR and UV/Vis regions under the effect of the annealing temperature. Besides the absorption band in the UV region, a strong and wide absorption band extending over the visible light region is observed. The active layer homogeneity appears in the absorbance, transmittance, and reflectance spectra and is related to the annealing temperature from 140 to 160 °C. Figure 1 shows nearly perfect transmission spectra with no evidence of active layer or thickness inhomogeneity or scattering. Furthermore, because it is practically difficult to measure these spectra at the same spot on the film multiple times, and as we carry out and produce the same spectra, the films must have good homogeneity in composition or thickness.

The effect of spinning frequency on the optical properties of the deposited P3HT:PC$_{60}$BM was studied, as shown in Figure 2a,b. The spinning frequency was varied from 900 to 3000 rpm. The P3HT:PC$_{60}$BM solution concentration was 2 wt.%, and the annealing temperature was 160 °C for 5 min. The transmission of the active layer is affected by the spinning frequency. In the IR region, the transmission is decreased visibly by increasing the spinning velocity, but in the UV/Vis region, a reversal behavior is observed. From Figure 2a,b, the optical properties confirm the strong absorption of the active layers in the UV and Vis range, especially from 300 to 650 nm. As the spinning velocity decreased, the optical absorption improved, which was related to the thickness effect of the active layer. Two absorption peaks are observed around 350 and 500 nm. The photoluminescence study of the prepared P3HT:PC$_{60}$BM is mentioned in Figure S1a,b (Supplementary Materials). From this figure, the optimum values for the PL study are observed for the films that were annealed at 160 °C for 5 min and deposited at 3000 rpm, which confirm the optimized values through the optical properties study.

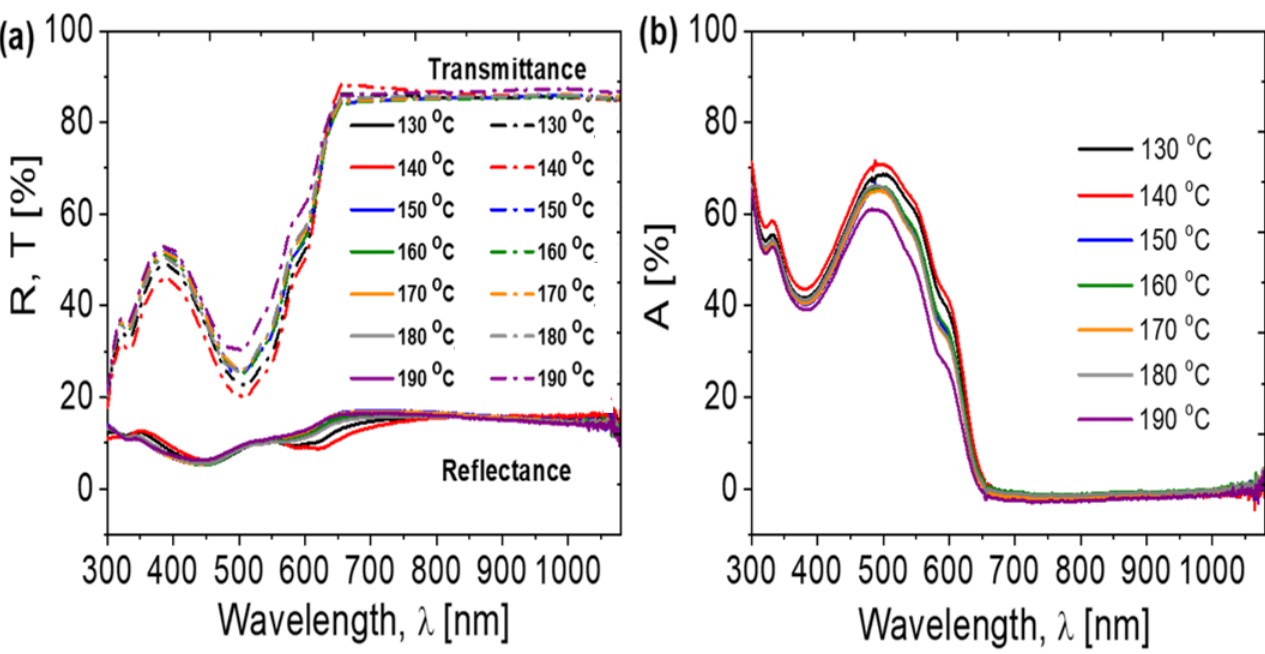

**Figure 1.** (**a**) Reflectance and transmittance and (**b**) absorbance spectra for P3HT:PC$_{60}$BM, which were prepared at 1500 rpm for 60 s and annealed at different temperatures.

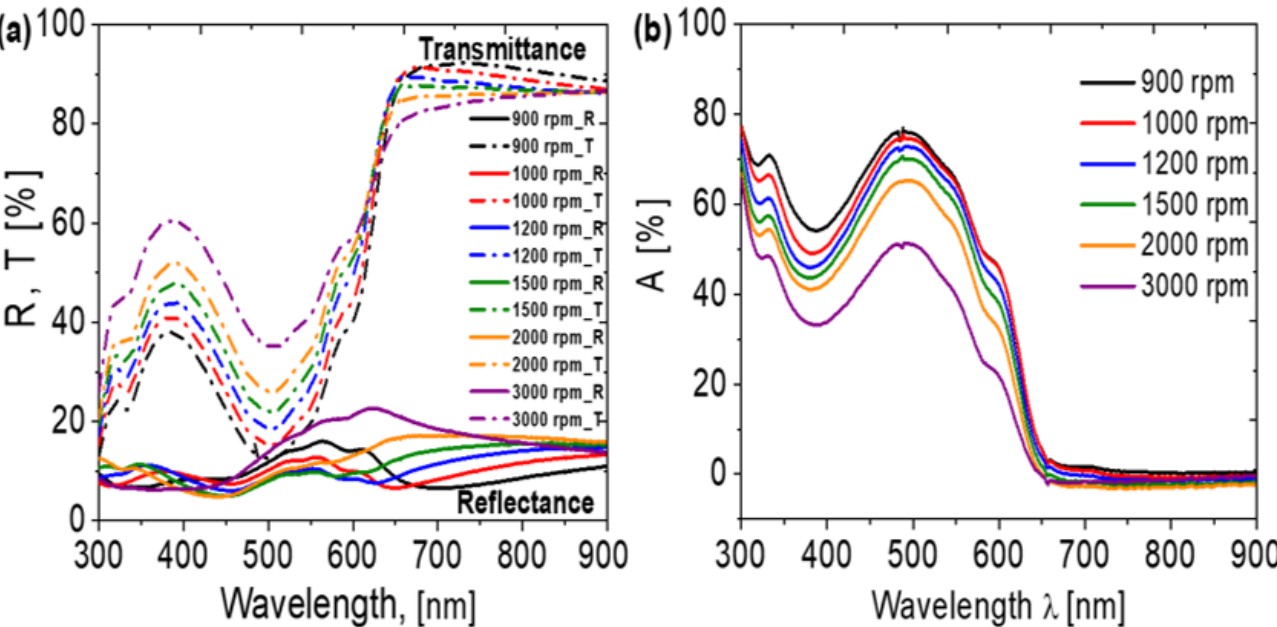

**Figure 2.** (**a**) Reflectance and transmittance, and (**b**) absorbance spectra for the active layers, P3HT:PC$_{60}$BM, which were prepared at different spin frequencies and annealed at 160 °C for 5 min.

The effect of post-production temperature from 130 to 190 °C on the J-V characteristics of the prepared OSCs under solar simulator and in the dark conditions are shown in Figure 3. The photovoltaic parameters (*PCE*%, *FF*%, $V_{OC}$, and $J_{SC}$), the parallel and series resistances ($R_P$, $R_S$) are estimated from the J-V characteristics in Figure 3 and presented in Figure 4a–f as a function of the annealing temperature. The values of the different parameters are reported in Table 1. The 160 °C annealing temperature has the optimum effect on the photovoltaic parameters of the prepared OSC, whereas the values of *FF*% and *PCE*% are 53% and 3.68%. Then, their values are decreased to 40% and 2.21% as the

temperature increased to 190 °C. The integral values for the I-V study under light are presented in Figure S2a.

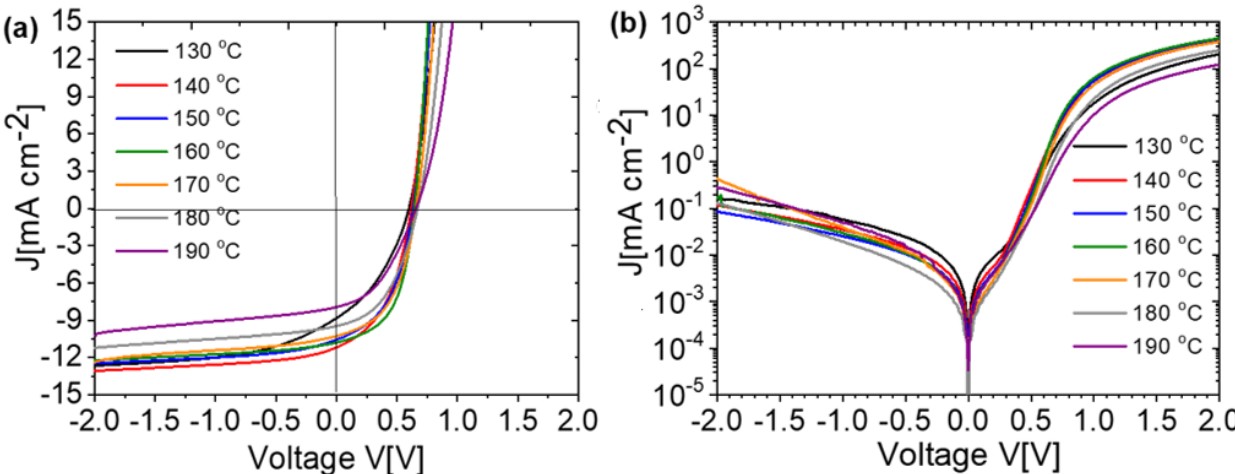

**Figure 3.** J-V curves under (**a**) the solar simulator and (**b**) in dark for the prepared OSC with the active layer prepared at different temperatures.

**Table 1.** Summary of the obtained photovoltaic parameters from the J-V data for OSCs with Al cathode, whereas the active layer was prepared at different annealing temperatures.

| Annealing Temperature (°C) | $J_{SC}$ (mA/cm²) | $V_{OC}$ (mV) | FF (%) | PCE (%) | $R_S$ (Ω) | $R_P$ (Ω) | Area (cm²) | $P_{max}$ (mW) | $V_m$ (mV) | $J_m$ (mA/cm²) |
|---|---|---|---|---|---|---|---|---|---|---|
| 130 | 8.82 | 597 | 37 | 1.95 | 8 | 376 | 0.42 | 0.82 | 360 | 5.43 |
| 140 | 11.17 | 616 | 45 | 3.09 | 6 | 579 | 0.42 | 1.3 | 400 | 7.74 |
| 150 | 10.56 | 631 | 47 | 3.15 | 5 | 651 | 0.42 | 1.32 | 430 | 7.32 |
| **160** | **10.78** | **640** | **53** | **3.68** | **5** | **1035** | **0.42** | **1.54** | **460** | **8** |
| 170 | 10.26 | 655 | 49 | 3.29 | 6 | 897 | 0.42 | 1.38 | 450 | 7.32 |
| 180 | 9.46 | 669 | 45 | 2.85 | 8 | 1055 | 0.42 | 1.2 | 420 | 6.79 |
| 190 | 8.24 | 664 | 40 | 2.21 | 13 | 987 | 0.42 | 0.93 | 390 | 5.67 |

Bold: Just to highlight the optimized OSC device at 160 °C.

The optimized OSC device at 160 °C illustrated $V_{OC}$ of 640 mV and $J_{SC}$ of 10.783 mA/cm². The values of $J_{SC}$ are decreased after 160 °C, as shown in Figure 4d. The small change in $J_{SC}$ at about 160 °C indicates that the increase in charge transportations is not accompanied by significant phase separation or breakdown in the active layer [34–36]. The increase in $J_{SC}$ values from 8.821 to 10.783 mA/cm² when increasing the annealing temperature from 130 to 160 °C refers to the improvement of the photon-to-current conversion efficiency (*EQE*). From the values of $V_{OC}$ and $J_{SC}$, the optimum power for the solar cell is obtained at 160 °C due to the multiply of the $V_{OC}$ and $J_{SC}$ values. Based on the obtained values of *PCE*, *FF*, $V_{OC}$, and $J_{SC}$; the optimum annealing temperature was 160 °C for the preparation of the OSC.

On the other hand, the J-V characteristics in dark at different annealing temperatures are illustrated in Figure 3b. From these characteristics, the estimated parallel resistance ($R_P$) increases by increasing the annealing temperature from 130 to 160 °C and then decreases as shown in Figure 4e. This highest $R_P$ value is obtained at 160 °C and contributes to the improvement of the vertical phase alignment. This indeed improves the performances of the OSCs under lighting. At the same time, there is a decrease in the $R_S$ value until it reaches 5 Ω at 150 and 160 °C, as shown in Figure 4f. The decrease of $R_S$ results in enhanced charge transportation, which also validates the observed increase of the fill factor (*FF*). In other words, the high *FF* at 160 °C of the optimized OSC resulted in a low $R_S$ of 5 Ω. This indicates the reduction of the potential barrier for holes' transportation at the interface.

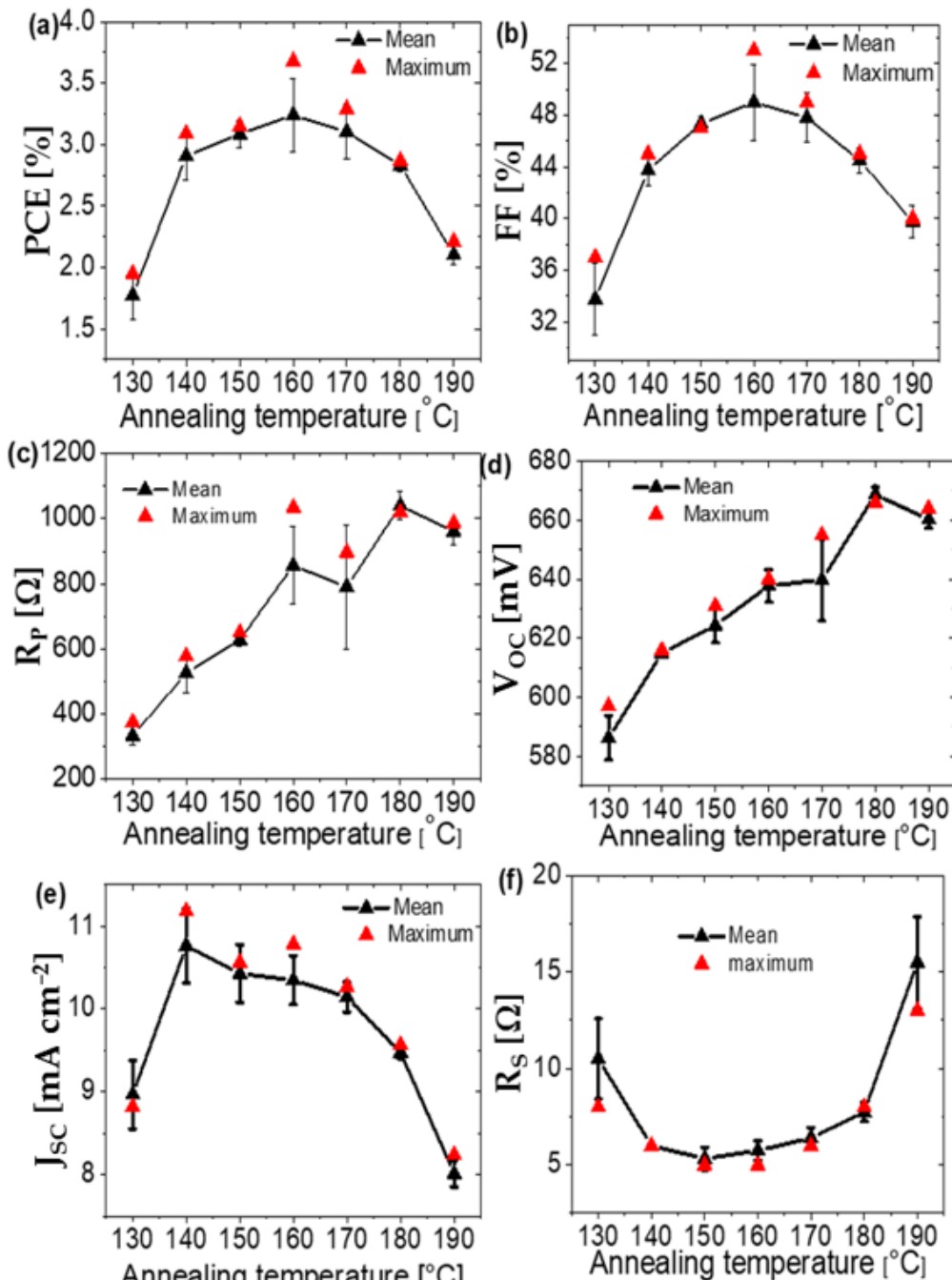

**Figure 4.** Variation of (**a**) *PCE*, (**b**) *FF*, (**c**) $V_{OC}$, (**d**) $J_{SC}$, (**e**) $R_P$, and (**f**) $R_S$ for OSCs with the active layer prepared at different temperatures from 130 to 190 °C.

To study the effect of the active layer thickness, P3HT:PC$_{60}$BM, on the efficiency of the prepared solar cell, the spin coating frequency is changed from 900 to 3000 rpm for this active layer coating. The concentration of the active layer solution is 2 wt.%, and the post-production annealing temperature is 160 °C for 5 min. In addition to that, the hole transport layer, PEDOT:PSS, spin frequency coating is 3000 rpm for 35 s, and the annealing temperature for this layer is 178 °C for 15 min. The study of different thicknesses was carried out on the prepared solar cell with Al and Mg-Al metals as cathodes.

Figure 5a,b represents the J-V curves under different spin coating from 900 to 3000 rpm for the active layer, P3HT:PC$_{60}$BM, under solar simulator and dark, respectively, using Al metal as a cathode. At the same time, Figure 5c,d represents J-V curves under different spin coating from 900 to 3000 rpm for the active layer, P3HT:PC$_{60}$BM, under solar simulator and

dark, respectively, using Mg-Al metal as a cathode. Table 2 summarized all the parameters obtained from Figure 5a–d. In addition to that, the statistical calculations for the different parameters are mentioned in Figures 6a–f and 7a–f. The high standard deviations may be due to the average data being taken from four-time repetition for four solar cells with the same preparation conditions. Moreover, the integral values for the I-V study under light are presented in Figure S2b,c (Supplementary Materials).

From Table 2 and the statistical calculations (Figure 6a,b), the optimum values of *PCE* are 4.21% at 2000 rpm and 4.65% at 3000 rpm for Al cathode and Mg-Al cathode, respectively. From Figure 5a and Table 2, the J-V characteristics under the solar simulator are affected by the spin coating frequency, 900 to 3000 rpm, of the deposited active layer. From Table 2, and the calculated statistical parameters in Figures 6 and 7, there is more enhancement in the parameters obtained using Al-cathode from J-V curves by increasing the spin coating frequency from 900 to 2000 rpm; $V_{OC}$ = 653 mV, *FF* = 54%, and *PCE* = 4.21%. The values of the optimum performance for the prepared OSCs are observed at 3000 rpm using the Mg-Al as a cathode; $J_{SC}$ = 12.01 mA·cm$^{-2}$, $V_{OC}$ = 660 mV, *FF* = 59%, and *PCE* = 4.65%. This means that there are more enhancements in the OSC performance by using Mg metal in the construction of the cathode. This Mg metal adds more enhancements in the electron transport from the active layer to the cathode.

Thus, an explanation can be introduced based on a diode equivalent circuit model containing two additional resistors $R_S$ and $R_P$ connected in series and parallel, as shown in Figure S3 (Supplementary Materials). Shockley equations for $V_{OC}$ and $J_{SC}$, can be used to describe current density-voltage curves and solve the circuit model analytically resulting in fitting parameters such as $R_S$, $R_P$, reverse saturation current density ($J_o$), and the ideality factor (*n*) of a solar cell that is similar to a single diode [37,38].

$$V_{OC} = \frac{KT}{e}\ln\left[1 + \frac{J_{ph}}{J_o}\left(\overset{\approx}{1} - \frac{V_{OC}}{J_{ph}\,R_P\,A}\right)\right] \simeq \frac{KT}{e}\ln\left[1 - \frac{J_{ph}}{J_o}\right], \tag{1}$$

$$J_{SC} = -\frac{1}{1 + \frac{R_S}{R_P}}\left[J_{ph} - J_o\left[\exp\left(\frac{|J_{SC}|R_S A}{\frac{nkT}{e}}\right) - 1\right]\right] \simeq -J_{ph}, \tag{2}$$

where $J_{ph}$ is the light-induced current density, *e* is the elementary charge, *k* is the Boltzmann constant ($8.617 \times 10^{-5}$ eV K$^{-1}$), *T* is the temperature, and *A* is the area of the cell. According to the equations above, a higher $R_P$ will likely increase $V_{OC}$, whereas a higher $R_S$ will likely limit $J_{SC}$. Thus, large $R_P$ and small $R_S$ are desirable. Equations (1) and (2) show that when $R_S$ is small and $R_P$ is high enough, parasitic resistance has a greater impact on *FF*. *FF* can be expressed as a function of normalized $V_{OC}$ ($V_{oc}$ = $eV_{OC}/nkT$), normalized $R_S$ ($R_S = R_S/R_{CH}$), and normalized $R_P$ ($R_P = R_P/R_{CH}$), where the characteristic resistance ($R_{CH}$) is defined as $V_{OC}/(J_{SC}A)$, and $R_{CH}$ represents the output resistance of the OSC at the maximum power.

The J-V curves in dark for the different spin coating frequencies, 900 to 3000 rpm, are illustrated in Figure 5b,d for the prepared OSC with Al and Mg-Al cathode, respectively. In addition, the statistical calculations of $R_P$ and $R_S$ for the dark current are mentioned in Figure 7c–f. From Table 2 and Figure 7c,d, the $R_P$ has values of 2139 and 2041 Ω (at 3000 rpm), for the prepared OSC with Al and Mg-Al cathode, respectively. On the other hand, the $R_S$ values have values of 7 and 5 Ω (at 3000 rpm), for Al and Mg-Al, respectively. For all electrochemical parameters calculated from J-V under the solar simulator or in dark, the Mg metal makes more enhancement in the electron transport in the prepared OSC for enhancing the efficiency.

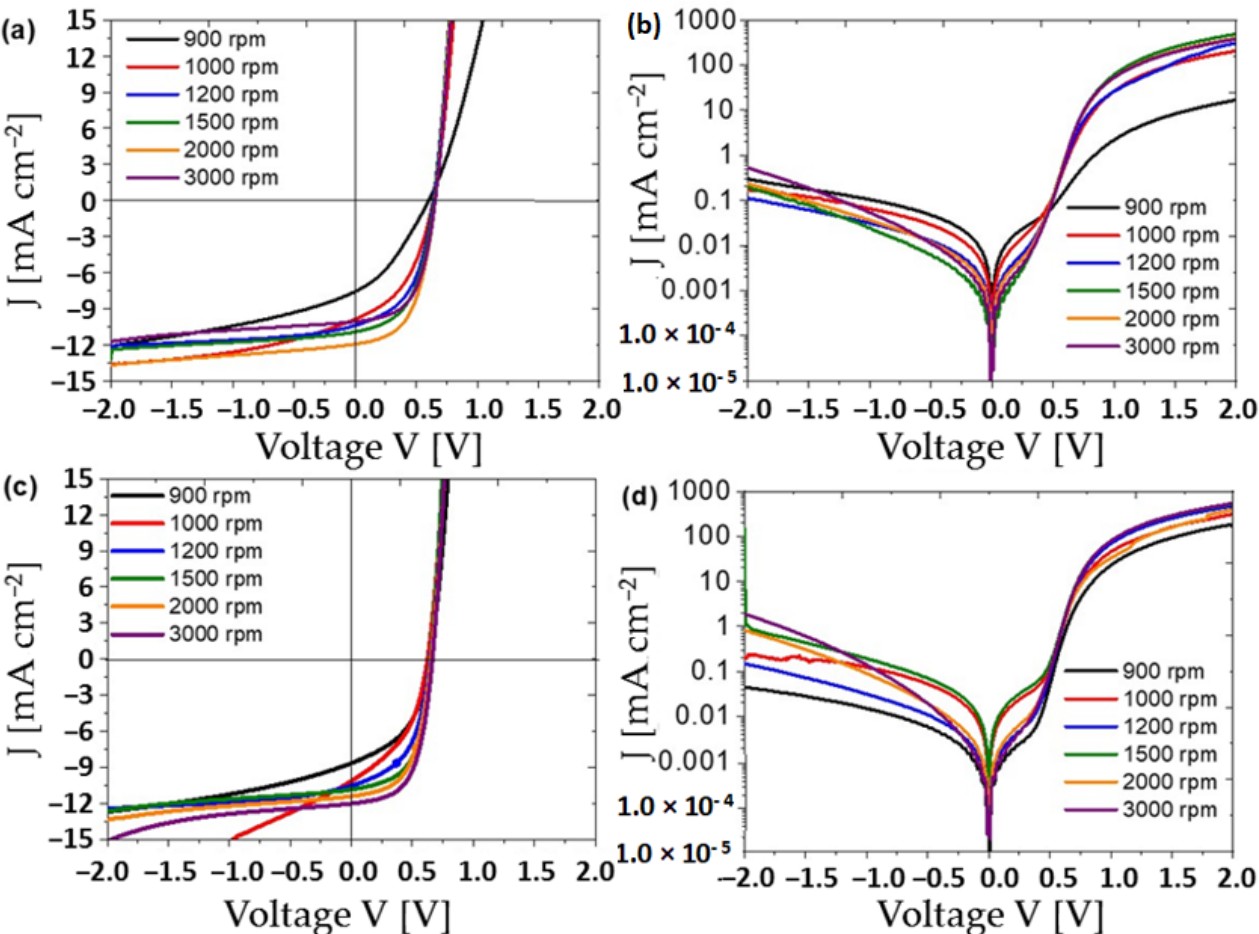

**Figure 5.** J-V curves under light (left column) and in dark (right column) for OSCs deposited at different spin coating frequencies from 900 to 3000 rpm with (**a**,**b**) Al cathode and (**c**,**d**) Mg-Al cathode.

**Table 2.** Summary of the obtained photovoltaic parameters from the J-V data for OSCs with Al and Mg-Al metal cathodes, whereas the active layer was prepared at different spin coating frequencies.

| Spin Frequency (rpm) | Cathode | $J_{SC}$ (mA/cm$^2$) | $V_{OC}$ (mV) | FF (%) | PCE (%) | $R_S$ ($\Omega$) | $R_P$ ($\Omega$) | $P_{max}$ (mW) | $V_m$ (mV) | $J_m$ (mA/cm$^2$) |
|---|---|---|---|---|---|---|---|---|---|---|
| 900 | Mg-Al | 8.649 | 639 | 47 | 2.61 | 9 | 599 | 1.1 | 450 | 5.80 |
| 1000 | Mg-Al | 10.138 | 620 | 44 | 2.74 | 5 | 380 | 1.15 | 430 | 6.38 |
| 1200 | Mg-Al | 10.524 | 642 | 52 | 3.49 | 5 | 754 | 1.47 | 460 | 7.59 |
| 1500 | Mg-Al | 10.844 | 646 | 58 | 4.09 | 4 | 1562 | 1.72 | 480 | 8.52 |
| 2000 | Mg-Al | 11.408 | 650 | 56 | 4.19 | 6 | 1755 | 1.76 | 470 | 8.91 |
| **3000** | **Mg-Al** | **12.01** | **660** | **59** | **4.65** | **5** | **2041** | **1.95** | **480** | **9.70** |
| 900 | Al | 7.571 | 602 | 34 | 1.55 | 5.7 | 477 | 0.65 | 330 | 4.71 |
| 1000 | Al | 9.872 | 636 | 44 | 2.78 | 8 | 536 | 1.17 | 410 | 6.79 |
| 1200 | Al | 10.397 | 637 | 50 | 3.32 | 5 | 784 | 1.39 | 450 | 7.37 |
| 1500 | Al | 10.929 | 644 | 52 | 3.67 | 6 | 1235 | 1.54 | 450 | 8.15 |
| 2000 | Al | 11.944 | 653 | 54 | 4.21 | 6 | 1591 | 1.77 | 460 | 9.15 |
| 3000 | Al | 10.08 | 653 | 56 | 3.71 | 7 | 2139 | 1.56 | 470 | 7.89 |

Bold: Just to highlight the results obtained for spin frequency of 3000 rpm.

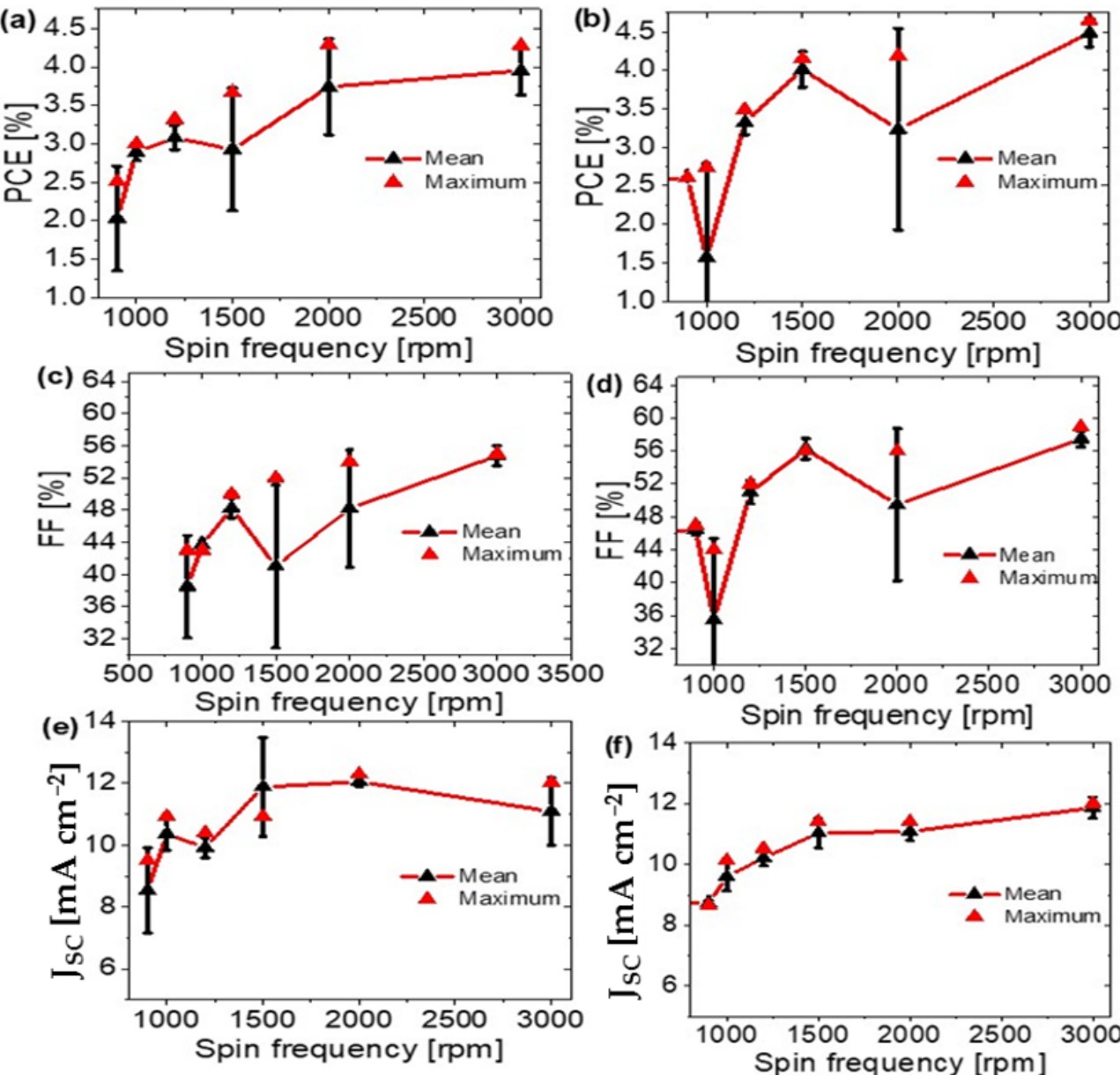

**Figure 6.** Statistical calculations of *PCE*, *FF*, and *J_{SC}* for the prepared OSCs under different spin coating frequencies with (**a**,**c**,**e**) Al and (**b**,**d**,**f**) Mg-Al cathode, respectively.

The external quantum efficiency (*EQE*) for the prepared OSC is determined under the effect of post-production annealing temperature, as shown in Figure 8. From this figure, high values of *EQE* are observed for the OSCs annealed at temperatures $\leq 160\ °C$. The highest *EQE* value is observed at 130 °C in the UV region, whereas in the visible light region the highest value is observed at 140 °C. Under the full UV-Vis region, 160 °C can be considered as the optimized annealing temperature for the designed OSCs. This is related to the response of the active layer to the incident photons and the generation of a high density of charge carriers.

To optimize the prepared organic solar cell, the *EQE* is determined at different thicknesses of the active layer and presented in Figure 9. From Figure 9a, the ITO/PEDOT:PSS/ P3HT:PC$_{60}$BM/Al OSC showed the highest *EQE* (59%) at 3000 rpm. Using Mg-Al cathode instead of Al cathode, Figure 9b, the *EQE* reached a value of 61% for ITO/PEDOT:PSS/ P3HT:PC$_{60}$BM/Mg-Al OSC. From the *EQE* values, the *Jsc* values were calculated and presented in Figures S4–S6 (Supplementary Materials).



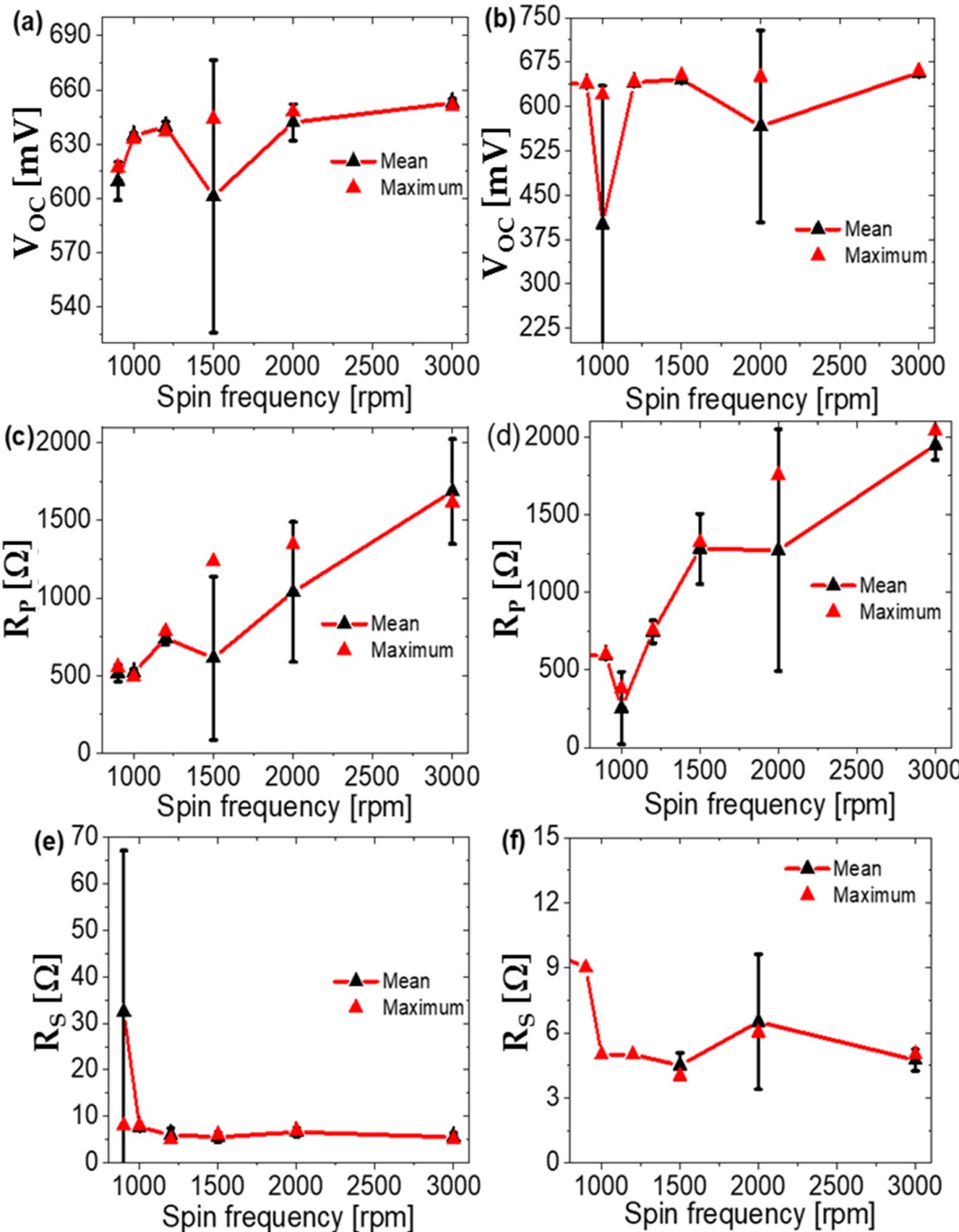

**Figure 7.** Statistical calculations of $V_{OC}$, $R_P$, and $R_S$ for the prepared OSCs under varied spin coating frequencies with (**a,c,e**) Al and (**b,d,f**) Mg-Al cathode, respectively.

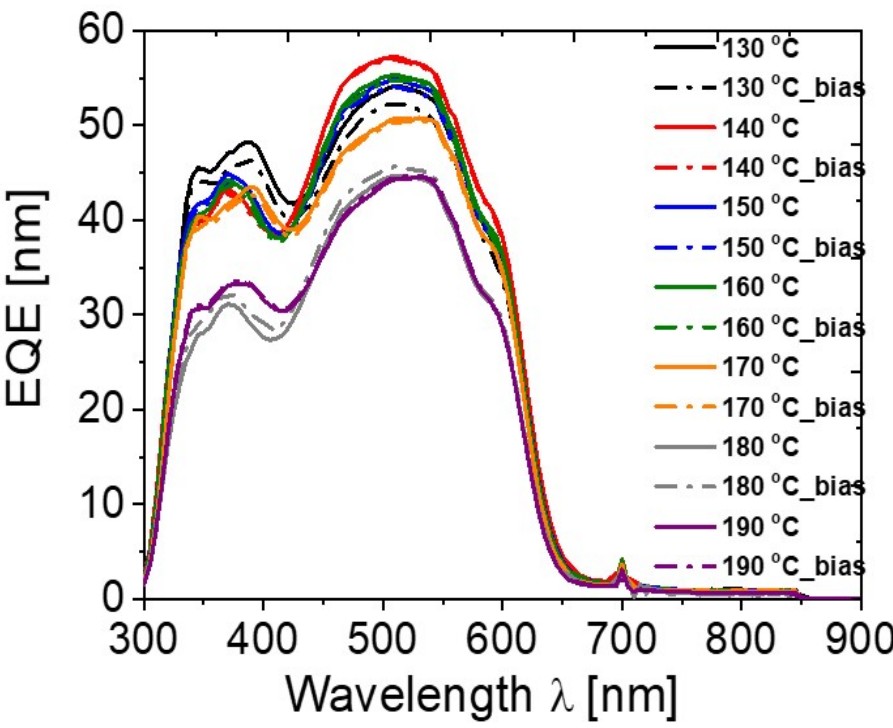

**Figure 8.** *EQE* efficiency of the prepared ITO/PEDOT:PSS/P3HT:PC$_{60}$BM/Al OSCs at varied post-annealing temperatures.

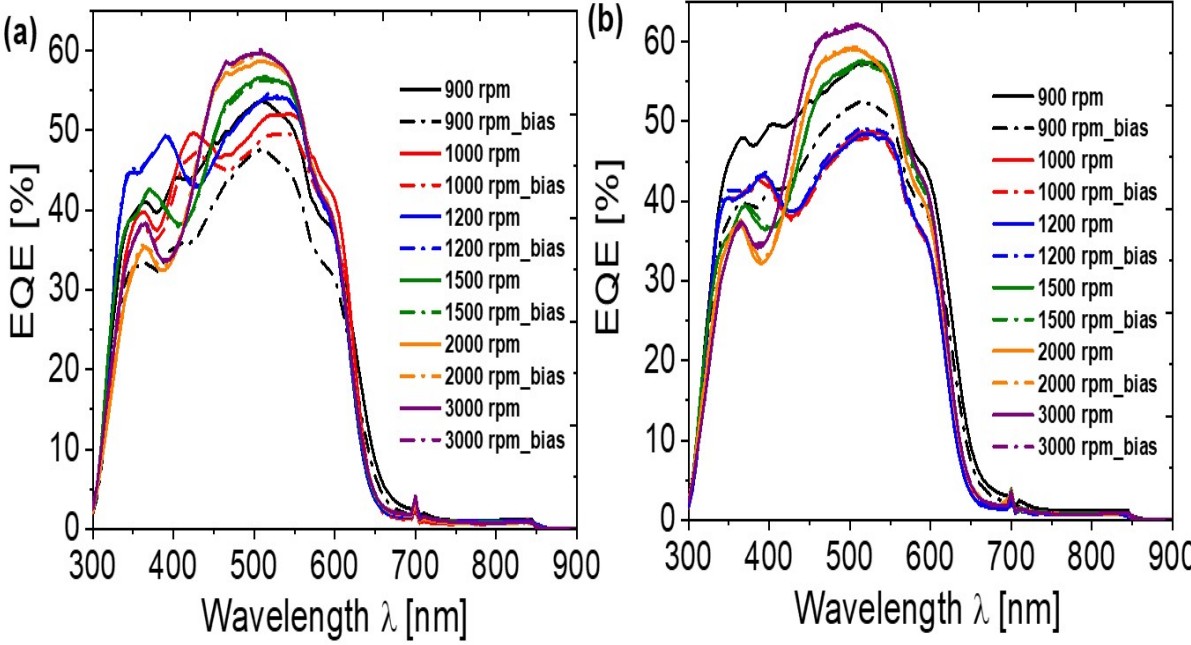

**Figure 9.** *EQE* efficiency of the prepared ITO/PEDOT:PSS/P3HT:PC$_{60}$BM/cathode OSCs at varied spin coating frequencies utilizing (**a**) Al and (**b**) Mg-Al as a cathode.

## 4. Conclusions

ITO-PEDOT:PSS/P3HT:PC$_{60}$BM/Al OSC is designed and optimized. The parameters of fabrication are optimized; post-production annealing temperature, P3HT:PC$_{60}$BM thickness, and cathode composition (Mg-Al). The HTL, PEDOT:PSS, was prepared under optimized conditions; 3000 rpm for 35 s and annealing temperature of 178 °C for 15 min. The active layers, P3HT and PC$_{60}$BM, with a concentration of 2 wt.% were deposited

under different spin frequencies, 900 to 3000 rpm, using the chlorobenzene as a solvent. The post-production annealing temperature was varied from 130 to 190 °C. For enhancing the electron contact between the active layer and cathode, a thin Mg metal layer is used with Al cathode. From our study, the optimum parameters were spin coating at 3000 rpm and an annealing temperature of 160 °C for 5 min, in the presence of Mg-Al as a cathode metal. The optimum J-V parameters values for the prepared solar cell were $J_{SC}$ = 12.01 mA/cm$^2$, $V_{OC}$ = 0.660 V, *FF* = 59%, *PCE* = 4.65%, and *EQE* = 61%. Therefore, we have synthesized a complete organic solar cell with acceptable efficiency using easy methods that can be implemented in the industry to provide a renewable energy source for people, particularly in remote areas. Furthermore, the anticipated cost of the synthesized solar cell is about 1 \$/cm$^2$, and we plan to cut the cost even more in the future by using more commercial materials.

**Supplementary Materials:** The following are available online at https://www.mdpi.com/article/10.3390/coatings11070863/s1, Figure S1: the photoluminescence of the P3HT:PCBM under (a) different annealing temperature, (b) different spin speed coating, Figure S2: The integral values of I-V values in light for the effect of (a) annealing temperature, (b) spin speed coating using Al electrode, (c) spin speed coating using Mg-Al electrode, Figure S3: The equivalent circuit model represents $R_S$ and $R_P$ in (a) dark and (b) light, Figure S4: The $J_{SC}$ and EQE relations under different wavelengths for (a) 130, (b) 140, (c) 150, (d) 160, (e) 170, (f) 180, and (g) 190 °C, Figure S5: The $J_{SC}$ and EQE relations under different wavelengths for (a) 900, (b) 1000, (c) 1200, (d) 1500, (e) 2000, and (f) 3000 rpm for the active layer using Mg-Al cathode, Figure S6: The $J_{SC}$ and EQE relations under different wavelengths for (a) 900, (b) 1000, (c) 1200, (d) 1500, (e) 2000, and (f) 3000 rpm for the active layer using Al cathode, Table S1: the $J_{SC}$ values calculated from EQE values from Figure S4 under different annealing temperatures for the active layer. Table S2. the $J_{SC}$ values calculated from EQE values from Figures S5 and S6 under different spin speed coating for the active layer.

**Author Contributions:** Data curation, M.S. and M.R.; Formal analysis, M.B.; Funding acquisition, M.S.; Investigation, M.B.; Methodology, A.A. and M.R.; Supervision, M.B. and A.A.; Writing—original draft, M.S. All authors have read and agreed to the published version of the manuscript.

**Funding:** This research was funded by Deanship of Scientific Research at Islamic University, Madinah, Saudi Arabia. Research project No.: 489.

**Institutional Review Board Statement:** Not applicable.

**Informed Consent Statement:** Not applicable.

**Data Availability Statement:** Not applicable.

**Acknowledgments:** We would like to thank the deanship of scientific research at the Islamic University of Madinah, Saudi Arabia for the financial support of this work through the program Tamayyuz II of the academic year 2020/2021, research project No.: 489.

**Conflicts of Interest:** The authors declare no conflict of interest.

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
