# Peer review of "Optimization of the Active Layer P3HT:PCBM for Organic Solar Cell"

_coatings, doi:10.3390/coatings11070863_

Round 1

Reviewer 1 Report

In the manuscript “Optimization of the Active Layer P3HT: PCBM for Organic 2 Solar Cell” the authors presented a study attempting to investigate the optimization of the active layer. The manuscript language and punctuations need to be revised again because some sentences are not understood. The main point of the study which to understand the optimization of the active layer on the efficiency of the Cell, however, the characterization techniques and the interpretations need to more improve.
1-    comment 1: the experimental measurements are not clear and need more description
2-    Comment 2: in the discussion and results, you only report the results without describing the scientific description.
3-    Comment 3: line 251, Figure S3?

Reviewer 2 Report

The manuscript presents interesting results related to organic solar cells. A wide variety of experiments were carried out to obtain solar cells of the highest possible efficiency. Technological parameters of the best cell were found. Almost all of my previous comments (this is not my first review of this paper) were included in the revised and resubmitted version of the manuscript. English was also improved. Some minor comments are as follow:

All variables should be italicized (lines 256, 257, 261, 263, …).

Remove ‘@’ in line 239.

What does ‘good Rp value’ mean (line 211)? ‘The highest’ could be more suitable in this context.

Something is missing in the caption of Fig. 6 (‘The average..’)

Correct the text in line 201 (‘by l be significant’? - unclear)

Round 2

Reviewer 1 Report

This paper is accepted for publishing in this journal.

This manuscript is a resubmission of an earlier submission. The following is a list of the peer review reports and author responses from that submission.

Round 1

Reviewer 1 Report

Following comments must be carefully addressed before further consideration.

(1) Novelty of the current work must be further clarified.

(2) Figure quality must be improved. In particular, texts are blurry.

(3) Figures 6 and 7: Why standard deviations are too big at certain points? Additional explanation is required.

(4) Current conclusions seem just a summary of the results. Give some insights of how the results can be applied.

(5) Abstrat: Also need to be modified accroding to my comment 4.

Reviewer 2 Report

The manuscript presents interesting results related to organic solar cells. A wide variety of measurements was carried out. However, I cannot recommend the manuscript for publication in this form. Some main concerns were listed below.  

The level of English is relatively low and must be improved. Some sentences have no scientific sense (i.e. ‘researchers are doing their best’, ‘varying spin coating’, ‘prepaid solar cell’, ‘active layer works well’ etc.). The article should be corrected by a native English speaker.

The experimental part should be more clear. There is no type of SEM. Moreover, there are also no results of SEM analysis in the next parts of the manuscript. Manufacturer of solar simulator could be provided. The methodology of statistical analysis presented in Figs. 6 and 7 should be described. What method was used for resistances Rs and Rp calculation?

Line(L) 128: It should be probably Figure 1b

L. 133-135: How the Authors recognize the homogeneity of the active layer from transmittance, absorbance, and reflectance spectra?

L. 143: ‘increasing the rpm’ – rpm is a unit, it would be better to use spinning velocity (or similar) instead of ‘rpm’

Table 1: Fig. 3 shows the I-V curves of solar cells fabricated at temperatures ranging from 130°C to 190°C. However, in Tab. 1 no parameters of I-V curves fabricated at 180°C and 190 °C were showed.

L. 159-160: did you mean ‘temperature decreased’ or increased? Verify and correct.

L. 202: ‘the optimum values of PCE are 3.71 and 4.65%’. According to Tab. 2 better current density was obtained for 2000 rpm and Al cathode (with PCE 4.21%). Why the optimum is 3.71 in this case (instead of 4.21 and 2000 rpm)? The same comment is for the lines from 207 to 208.

L. 208: It should be PCE instead of PEC.

Figs. 6-7: The main goal of the presented research is to optimize some of the technological parameters. However, in some cases, the differences in parameters are very high. For example, in Fig. 6a PCE for 1500 rpm or 200 rpm changes between 2 % and almost 4 %. What is the reason? It seems that you have an unstable (unpredictable) technology. Or perhaps, there is another reason.

There is no comparison of the obtained results with other researches. I would recommend adding the discussion part of the presented results comparing with other papers in the field.

Reviewer 3 Report

Please find the attachment for the details of the comments.

Round 2

Reviewer 1 Report

The authors adequately addressed the reviewer's comments.

Reviewer 2 Report

The manuscript has been revised according to most of my previous comments. However, English is still not good enough (‘Also’, ‘So’ in Abstract should be avoided). The article should be corrected by a native English speaker. The new part of the text (in lines 260 - 280) is rather a theoretical description than a discussion of results. Please, consider moving this part to the Introduction. Moreover, consider rewriting (with detailed explanation) the following sentence ‘four-time repetition for four solar cells with the same preparation conditions’ (245-246) in the Methodology section. More comments were listed below.

  • All variables should be italicized.
  • It is not common to use ‘pointers’ to figures from supplementary materials (Figure S2 etc.).
  • Remove ‘@’ in lines 127, 249.
  • Use for example ‘spinning velocity’ instead of ‘rpm’ in line 169
  • The same format of Vm values could be used in Tab. 1: (i.e. 419.9 -> 420, etc.)

Reviewer 3 Report

The manuscript has been significantly improved. However, please check the following.

  1. Please cross-check all the reference once again as it seems there are some mistakes in citing the work. For Example, in ref [37], 2021 appeared two times. Also, there is no uniformity in the references. It is suggested to use referencing software (EndNote, Mendeley etc) and make sure to cite the references according the journal's desired style.
  2.  I am not sure does coating journal allows supplementary information or not. Please check, if it does not allow, it is better to use supplementary information in the main manuscript at respective places.
  3. The explanation covering diode equivalent circuit model can be further improved.
  4. It is suggested to use PC60BM everywhere in the manuscript instead of writing only PCBM. It will improve the clarity to the reader. 
  5. It is suggested to include 'parasitic resistances' in the keywords.
